# Strengthening the Evidence-Based Approach to Guiding Effective Influenza Vaccination Policies

**DOI:** 10.3390/vaccines8030342

**Published:** 2020-06-27

**Authors:** Giovanna Elisa Calabrò, Maria Lucia Specchia, Sara Boccalini, Donatella Panatto, Caterina Rizzo, Stefano Merler, Anna Maria Ferriero, Maria Luisa Di Pietro, Paolo Bonanni, Chiara de Waure

**Affiliations:** 1Section of Hygiene, Department of Life Sciences and Public Health, Università Cattolica del Sacro Cuore, 00168 Rome, Italy; alisacalabro@icloud.com (G.E.C.); marialuisa.dipietro@unicatt.it (M.L.D.P.); 2Value in Health Technology and Academy for Leadership & Innovation (V.I.H.T.A.L.I.), Spin-off of Università Cattolica del Sacro Cuore, 00168 Rome, Italy; 3Fondazione Policlinico Universitario Agostino Gemelli IRCCS, 00168 Rome, Italy; 4Department of Health Sciences, University of Florence, 50134 Florence, Italy; sara.boccalini@unifi.it (S.B.); paolo.bonanni@unifi.it (P.B.); 5Department of Health Sciences, University of Genoa, 16132 Genoa, Italy; panatto@unige.it; 6Predictive and Preventive Medicine Research Unit, Multifactorial and Complex Disease Research Area, Bambino Gesù Children’s Hospital, IRCCS, 00165 Rome, Italy; caterina1.rizzo@opbg.net; 7Bruno Kessler Foundation, 38122 Trento, Italy; merler@fbk.eu; 8Directorate-General for Health Planning, Ministry of Health, 00144 Rome, Italy; annamariaferriero@hotmail.it; 9Department of Experimental Medicine, University of Perugia, 06132 Perugia, Italy; chiara.dewaure@unipg.it

**Keywords:** influenza, Health Technology Assessment, recommendations, vaccines

## Abstract

The availability of several effective and safe vaccines enables health systems to counteract annual influenza epidemics. However, the criteria of appropriateness and sustainability require that each citizen should receive the right vaccine. The value of each vaccine can be assessed within well-known frameworks, such as the Health Technology Assessment (HTA), a step that is fundamental to the process of allocating resources to vaccination strategies. The paper describes how HTA has been incorporated as an evidence-based tool to support the definition of Italian vaccination strategies, reports the results of the HTA report on the most recently available influenza vaccine in Italy (cell-based quadrivalent vaccine (QIVc)—Flucelvax^®^ Tetra) and elaborates on current and future recommendations in the field of influenza vaccination. Recommendations issued by the Italian Ministry of Health foster the appropriate use of influenza vaccines from 2018–2019 onwards. Evidence of the value of newly available vaccines will hopefully support future decisions and promote the appropriate use of these vaccines on the basis of the characteristics of the target population. However, the success of influenza vaccination will also depend on citizens’ empowerment and engagement in the decision-making process.

## 1. Introduction

Vaccinations prevent around 2.5 million deaths every year and protect both children and adults against the threat of vaccine preventable diseases (VPDs). As part of a set of public health interventions for disease prevention and control, vaccines and immunization strategies are a fundamental investment in the future of a country and of the whole world [1].

In the field of vaccination, particular attention must be paid to influenza, since seasonal influenza constitutes a serious public health problem, and has a considerable epidemiological, clinical and economic impact [2]. This is attributable to several factors, such as: the ubiquity and contagiousness of the disease, the antigenic variability of the virus, the epidemic (and sometimes pandemic) and seasonal trend, the possibility of serious complications, especially in some categories of subjects (children, the elderly, people with comorbidities and chronic diseases), the costs of managing influenza-related complications and social costs (working days lost, loss of productivity, etc.). Indeed, influenza imposes a significant burden on health, the healthcare system and society. The Burden of Communicable Diseases in Europe (BCoDE) study has attributed 30% of the total burden to influenza, due to 31 selected infectious diseases for a total of 81.8 DALYs per 100,000 of the population each year [3]. As for the economic impact, there is a plenty of evidence that provides different results, depending on considered drivers and sources of data. As for Italy, the average cost of an influenza season has been estimated EUR 13 billion, considering the cost for single case of EUR 212, EUR 731 and EUR 1041 in the pediatric, adult and elderly population, respectively [4]. With respect to working population, working days lost due to influenza add up to 11,100 per year with a cost of EUR 327 per person [5]. 

Influenza vaccination is able to reduce the utilization of healthcare services in all ages [6,7,8], and has been shown cost-saving or cost-effective in the most of available evidence [9]. Nonetheless, despite international and national recommendations on influenza vaccination, and the availability of a broad spectrum of safe, effective and cost-effective vaccines, vaccination coverage levels, both in Italy and in Europe, remain rather low. Successful preventive strategies should be therefore supported and implemented. In this light, it is particularly important to improve the management of the available influenza vaccines by assigning the most suitable vaccine to each population group [10]. The principle of vaccination appropriateness must therefore be applied, in order to guarantee every citizen adequate protection [10]. In this regard, the Health Technology Assessment (HTA) constitutes a rigorous method of producing evidence to support the definition and implementation of appropriate and effective vaccination strategies [11]. 

The objectives of this paper are to: (1) describe how HTA has been incorporated into the evidence-based approach to the definition of Italian vaccination strategies; (2) briefly report the results of the HTA of the most recent influenza vaccine made available in Italy (cell-based quadrivalent vaccine (QIVc)—Flucelvax^®^ Tetra (Seqirus, Inc., Summit, NJ, USA)); (3) depict current and future recommendations in the field of influenza vaccination.

## 2. HTA as Evidence-Based Approach to the Definition of Italian Vaccination Strategies

The 2012–2014 Italian National Immunization Programme (NIP) was the first to define the criteria for the assessment of new and currently used vaccines for inclusion in the national immunization program [12]. These criteria, which are still being applied, refer to the HTA approach, namely “a multidisciplinary process that summarizes information about the medical, social, economic and ethical issues related to the use of a health technology in a systematic, transparent, unbiased, robust manner. Its aim is to inform the formulation of safe, effective, health policies that are patient focused and seek to achieve best value” [13].

Indeed, for every vaccine, the 2012–2014 NIP underlined the importance of assessing the epidemiological burden of the VPD, the efficacy and safety of the vaccine under consideration and any alternatives. With regard to new vaccines, a forecast of their health and economic impact is required, together with an evaluation of their organizational, ethical, social and legal implications. 

In accordance with this approach, several influenza vaccines have been assessed through the HTA in Italy [14,15,16] and in 2018 a National Immunization Technical Advisory Group (NITAG) has been appointed and entrusted to adopt the HTA approach, in order to release evidence on vaccination strategies. 

Furthermore, the HTA was recognized as fundamental tool for the future of vaccination in Italy, because it makes it possible to fulfil one of the ten conditions laid down by the 2017–2019 NIP, namely investment [17]. Indeed, even though health technologies may lead to cost savings and health benefits in the short and/or long term, they require immediate investment. 

However, the success of a vaccination strategy depends upon the actions of several stakeholders, citizens among them. In fact, in the very end, the citizen decides whether he/she will adhere or not to the vaccination. Therefore, citizens are expected to participate in the decision-making process through involvement in patient advocacy groups, advisory boards and HTA bodies [11]. They should also eventually be engaged and empowered in order to make a proper use of technologies.

## 3. The HTA as an Evidence-Based Tool for the Introduction of New Vaccines in the Italian Healthcare Context: Overview of the HTA Report on the Most Recent Influenza Vaccine (Cell-Based Quadrivalent Vaccine (QIVc)—Flucelvax^®^ Tetra) Made Available in Italy

The production of an HTA report and its dissemination at national level can support decision-makers and foster the evidence-based introduction of a new health technology in the healthcare setting. Nevertheless, the scientific dissemination of HTA reports also deserves to be expanded beyond national contexts, to facilitate comparisons between different countries on the introduction of new health technologies. In this light, considering also the principles stated in the 2012–2014 NIP and the subsequent 2017–2019 NIP, we have summarized below the main findings of the Italian HTA report on the new cell-based quadrivalent vaccine (QIVc) [15]. 

In accordance with the EUnetHTA core model, the first aspect addressed in the HTA report was the health problem and its current management. This involved conducting a systematic review of the available literature on the burden of disease, in terms of both complications and mortality, and of current alternatives to the cell-based vaccine, namely the adjuvanted trivalent vaccine (aTIV) and the non-adjuvanted egg-based quadrivalent vaccine (QIVe). Furthermore, data from the Italian epidemiological and virological surveillance system were collected and analyzed. 

Over nine influenza seasons, the cumulative incidence of influenza-like illnesses (ILI) was estimated as 10.1%. Marked differences were observed among age-groups, the cumulative incidence of ILI being four to nine times higher in children 0 to 4 years of age than in people aged 65 years or older. Type A influenza virus circulated more frequently than type B, though the latter has become increasingly frequent in recent years, particularly in children. Regarding the two B lineages, Yamagata was the more common and was also associated with more severe cases. By contrast, older people were mostly affected by type A H3N2 infection. Hospitalization rates were seen to have declined over the years, with the highest rates among people aged 75 years or older. Nevertheless, the data on the burden of influenza revealed complication rates as high as 60% among the elderly and subjects with comorbidities. Children also suffered complications, particularly otitis and acute bronchitis. While the overall mortality rate was under two per 10,000 inhabitants, excess mortality was estimated as 1.9–2.2 per 100,000 on considering only influenza and pneumonia. The available influenza vaccines proved to be safe and immunogenic. The aTIV was shown to be more immunogenic than the non-adjuvanted vaccines and to have an effectiveness of 58.6% (95% CI: 24.3–77.4%) against influenza and 51% against hospitalization for influenza and pneumonia. The QIVe vaccine was 59% effective against influenza, though data regarding protection against hospitalizations were more controversial. This may also be ascribed to the smaller amount of evidence available for QIVe, which has been used since the 2014/2015 season, whereas aTIV has been in use since 1997.

The HTA report also examined data on the immunogenicity, efficacy and effectiveness of QIVc. Obviously, these data were scant and limited to one season, but they pointed to a potential additional benefit of the new vaccine in comparison with the others, with respect to both influenza and hospitalizations.

The second aspect assessed in the HTA report was the modelling of the adoption of the QIVc, taking into account both the dynamics and the burden of infection. In this regard, the model of susceptible (S), infected (I) and recovered (R) was considered, together with a decision tree, reproducing the natural history of the infection. The model predicted the health and economic impact of replacing QIVe with QIVc in the 9–74-year age-group in terms of Euro per quality adjusted life years (QALY) gained. This scenario was chosen on the basis of the authorization received from the European Medicines Agency, and on the Italian MoH recommendation of the preferential use of aTIV in subjects aged 75 years or older. Modelling was performed on considering one season and both direct and indirect costs. In the base-case scenario, replacing QIVe with QIVc was cost-effective on considering a threshold of 30,000 EUR/QALY. This result was also confirmed in the analysis of different scenarios and in the probabilistic analysis. 

Finally, the HTA also dealt with the organizational and ethical implications of adopting the new technology. The organizational aspect was investigated through a narrative review of the available evidence, while ethical concerns were examined within a shared international framework. This analysis revealed the importance of promoting an appropriate use of vaccines, in order to achieve the goal of value-based healthcare. 

## 4. Current and Future Recommendations on Influenza Vaccination

Some important steps have already been taken towards improving influenza vaccination appropriateness in Italy. During the 2018–2019 and 2019–2020 influenza season, the use of aTIV was recommended by the Italian MoH for people over 75 years of age [18,19]. This recommendation was based on two observations: the fact that the highest burden of the H3N2 virus was borne by the elderly, and evidence of the improved efficacy of the adjuvanted formulation in this age-group; consequently, aTIV could be expected to provide greater protection than the trivalent or non-adjuvanted quadrivalent vaccines in this category.

Today, several vaccines are available for influenza, but the availability of new vaccination technologies must be always assessed, in order to maximize results in terms of health and guarantee the community adequate protection against influenza. Therefore, the evidence provided by our HTA report on QIVc may be hopefully helpful in drawing up and issuing the new MoH’s recommendations on influenza vaccination. In particular, it is expected that the MoH provides healthcare professionals with instructions on the preferential use of each vaccine in the different target groups.

In fact, the choice of a preventive, diagnostic, therapeutic or rehabilitative health service must necessarily meet the criteria of effectiveness, efficiency, safety and appropriateness, as these are essential dimensions of healthcare quality [20,21]. More specifically, the criterion of appropriateness, given its complexity and multidimensionality, is closely related to all the others, and has acquired a decisive role in the international health policy landscape during the years [22]. The first definitions emphasized the risk-benefit ratio, whereby a procedure was deemed "appropriate" if its expected benefit exceeded any negative consequences by a sufficiently large margin. Over time, however, economic concerns made it necessary to also consider the financial sustainability of health technologies [22]. 

The choice of a vaccine also needs to respond to the requirement of appropriateness, which must guide the use of the available products according to the characteristics and health needs of the various age-classes and specific population groups [10]. Among the available options that could reduce the significant burden of influenza, further prospects are offered by the new QIVc, which is produced by means of an alternative platform based on cell culture [23,24], and is expected to have a higher correspondence with circulating viral strains than vaccines obtained by means of the traditional egg-based production method, thereby offering better protection against infection [25]. In this regard, it is also interesting to remember that, during the consultation and information meeting on the composition of influenza vaccines to be used in 2019, the WHO recommended changing the H3N2 virus produced in eggs (A/Switzerland/8060/2017 (H3N2)-like virus), but not that produced in cells (A/Singapore/INFIMH-16-0019/2016-like virus) [26]. This decision, together with the emerging data on the vaccines produced through cell cultures [25], constitutes a further encouraging perspective on the effectiveness of this new vaccine. All these considerations should be borne in mind when allocating resources in the next influenza vaccination campaigns, even though it will also be crucial to support initiatives for citizens’ empowerment and engagement in the decision-making process.

## 5. Conclusions

HTA is an essential evidence-based approach for evaluating new vaccines. Evidence of the value of newly available vaccines is indeed crucial to support decisions on their introduction and reimbursement within the healthcare setting and to promote their appropriate use. This is especially true for influenza vaccination as different vaccines are available and various target groups require to be vaccinated. Available HTA projects on influenza vaccines could assist decision-makers in formulating recommendations for the seasonal influenza vaccination also considering citizen involvement.

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
