# Peer review of "Strengthening the Evidence-Based Approach to Guiding Effective Influenza Vaccination Policies"

_vaccines, 2020, doi:10.3390/vaccines8030342_

Round 1

Reviewer 1 Report

The communication described how HTA has been incorporated into the agenda of the Italian vaccination policy and provided some recommendations according to the results of the HTA report. This paper is well written and provides a clear objective and presentation of messages. Below are a few comments:

  1. The acronym, VPDs, may spell out the full name (“Vaccine Preventable Diseases”).
  2. The author(s) might provide more evidence on the costs of managing influenza-related complications and social costs (Page 2, Line 52-53). For example, how much healthcare expenditure can be saved if people receive a flu vaccination? It might illustrate the significance of vaccination for people and healthcare systems.
  3. Did the HTA report compare aTIV and QIVc? The author(s) mentioned that “Moreover, the evidence yielded by our HTA report on QIVc may be of great importance in drawing up and issuing the MoH's next recommendations on influenza vaccination.” (Page 4, Line171-172) Does it imply that QIVc is better than aTIV for older adults over 75 years of age? It might need to clarify.

Author Response

We thank the Reviewer for the general comment and his/her suggestions that have given us the possibility to improve the quality of the manuscript. Hereafter, point-to-point answers have been reported.

Below are a few comments:

  1. The acronym, VPDs, may spell out the full name (“Vaccine Preventable Diseases”).

Thank you for your warning, we have indicated the full name “Vaccine Preventable Diseases”.

  1. The author(s) might provide more evidence on the costs of managing influenza-related complications and social costs (Page 2, Line 52-53). For example, how much healthcare expenditure can be saved if people receive a flu vaccination? It might illustrate the significance of vaccination for people and healthcare systems.

Thank you for your suggestion, we have provided more evidence to better illustrate the impact of influenza on health, healthcare systems and society. In particular, we have reported data on DALYs attributable to influenza and average costs of an influenza season.

  1. Did the HTA report compare aTIV and QIVc? The author(s) mentioned that “Moreover, the evidence yielded by our HTA report on QIVc may be of great importance in drawing up and issuing the MoH's next recommendations on influenza vaccination.” (Page 4, Line171-172) Does it imply that QIVc is better than aTIV for older adults over 75 years of age? It might need to clarify.

Thank you for your request. As reported in the paper “The model predicted the health and economic impact of replacing QIVe with QIVc in the 9 - 74-year age-group in terms of Euro per quality adjusted life years (QALY) gained. This scenario was chosen on the basis of the authorization received from the European Medicines Agency and on the Italian MoH recommendation of the preferential use of aTIV in subjects aged 75 years or older”. As for the potential impact of the results of our report on future recommendations, we did not refer to the replace of aTIV with QIVc, even though the latter may be also used in the elderly. We have included the following sentence to better address the expected impact “In particular, it is expected that the MoH provides healthcare professionals with instructions on the preferential use of each vaccine in the different target groups”.

Reviewer 2 Report

In this paper, the authors describe how HTA has been incorporated into the agenda of Italian vaccination policy and detail the relevant recommendations for the use of influenza vaccines from 2018 onwards. The author put emphasis on the concept of “appropriateness” for the choice of influenza vaccine, tailored on the population target. The work reviews previous works and policy documents, with a very valuable objective. That said, there are several issues that should be revised before planning its publication:

(note: the numbers refer to the lines #, as received in the submitted pdf file)

Content:

Page 1 and 2: the introductory part of 1.5 pages seems unbalanced in view of the total length of the article (less than 4 pages, if excluding the abstract, disclosures and bibliography). Furthermore, such introduction restates very basic information widely available, and extensively discussed in past works. While this could be appreciable to read in an detailed review paper, it seems a bit too extensive for a short communication, especially given the redundancy with past publications of the same or different research groups.

  1. “The objectives of this paper are to: 1) describe how HTA has been incorporated into the agenda of Italian vaccination policy”. Such description was expected in the paragraph titled “2. Incorporation of HTA into Italian vaccination policy” (l.96-113). Unfortunately, aside of succinctly citing previous works and providing a a vague overview by stating generalities, this part of the work does not provide any new valuable information that has not been already published. An in-depth and detailed description of how HTA has been technically incorporated in the Ministry of Health decisions should be included in order to fullfill the expectations given by the abstract and by the statement on line "92". This is the main promise of this manuscript, and it totally misses its goal by remaining very vague.

114-163: while it is appreciable to read a summary of a previously published report, it is unclear what new information it brings, and how it would justify a new publication.

177-178: it is appreciated that the authors that the concept of “appropriateness” is far from being new.

183-192: the authors interestingly develop the issue of egg-adaptation, to explain one of the factors that led to the choice of QIVc vaccine for the majority of the population (except for 75+ yo). That said, was it useful to cite that in a policy-oriented manuscript, instead of focusing on the results?

193-197: how did this information directly impact the HTA, and later on, the Guidelines for vaccination in Italy?

201: it is unclear why SARS-CoV-2 is cited here, since there is no available vaccine yet, and the decision-making in regard to vaccination in regard will probably be influenced by dynamics very different from traditional HTA evaluation, once available. It is strongly encouraged to avoid references to topics that are unrelated, and thus unnecessary.

Nature of submission: this article has been submitted as “Communication” (at the best of my knowlegde), but does not match the requirement of “preliminary, but significant, results” as defined by Vaccines, since this manuscripts mostly summarizes previously published works and policies, that are already in place.

Language and editing:

  1. “keep ongoing”?.
  2. “to guiding”? To guide or guiding? For effective guidance?

Authors should revise the title in its entirety.

27: allocation process of what? Of fundings? Of the vaccine to patients?

  1. VPD: please specify acronym
  2. Indeed? It is unclear what authors mean
  3. check tabulation

The choice of English translations for some words - while not depriced of sense -  could be revised with some more commonly used terms to improve the understanding: i.e. repercussion (translated from “repercussioni”) vs impact; appropriateness (translated form “appropriatezza”) vs suitability.

Furthermore, there are many sentences that the authors may want to rephrase in order to increase the comfort and understanding of the readers.

Author Response

We thank the Reviewer for the general comment and her/his suggestions that have given us the possibility to improve the quality of the manuscript. Hereafter, point-to-point answers have been reported.

  1. Page 1 and 2: the introductory part of 1.5 pages seems unbalanced in view of the total length of the article (less than 4 pages, if excluding the abstract, disclosures and bibliography). Furthermore, such introduction restates very basic information widely available, and extensively discussed in past works. While this could be appreciable to read in a detailed review paper, it seems a bit too extensive for a short communication, especially given the redundancy with past publications of the same or different research groups.

Thank you for your suggestion, we have reduced the introductory part to balance the total length of the article. Nonetheless, following the suggestions from Reviewer 1 we have also included some data on the burden of influenza.

  1. “The objectives of this paper are to: 1) describe how HTA has been incorporated into the agenda of Italian vaccination policy”. Such description was expected in the paragraph titled “2. Incorporation of HTA into Italian vaccination policy” (l.96-113). Unfortunately, aside of succinctly citing previous works and providing a vague overview by stating generalities, this part of the work does not provide any new valuable information that has not been already published. An in-depth and detailed description of how HTA has been technically incorporated in the Ministry of Health decisions should be included in order to fulfil the expectations given by the abstract and by the statement on line "92". This is the main promise of this manuscript, and it totally misses its goal by remaining very vague.

Thank you for your observation. It is well known that the assessment is followed by the appraisal which is commonly relied on a more participative approach that lacks standardization. Indeed, it is quite hard to fairly and thoroughly describe how the HTA is eventually incorporated in MoH decisions. Consequently, we have modified the aim of the paper with respect to this point and the title of the relevant paragraph. In particular, we have focused the attention on HTA as an evidence-based approach to the definition of vaccination strategies. Furthermore, in order to expand on the role of HTA in the definition of vaccination strategies in Italy, we have also included a reference to the establishment and the role of NITAG.

  1. 114-163: while it is appreciable to read a summary of a previously published report, it is unclear what new information it brings, and how it would justify a new publication.

Thank you for your observation. We have better explained that reporting a summary of the published report is useful to address the role of HTA as an evidence-based tool for the evaluation of new vaccines in the Italian care context (we have also modified the title of the paragraph accordingly). In this regard, we have reported the evidence about the new vaccine that entered the market during the last influenza season. In order to answer to your comment, we have changed the paragraph title and included further details to justify this new publication. In particular, we have pointed out that the scientific dissemination of HTA reports deserves to be expanded beyond national contexts to facilitate comparisons between different countries on the introduction of new health technologies.

  1. 177-178: it is appreciated that the authors that the concept of “appropriateness” is far from being new

Thank you for your comment, we have removed the reference to eighties’.

  1. 183-192: the authors interestingly develop the issue of egg-adaptation, to explain one of the factors that led to the choice of QIVc vaccine for the majority of the population (except for 75+ yo). That said, was it useful to cite that in a policy-oriented manuscript, instead of focusing on the results?

Thank you for your observation, we agree with you and we have consequently reduced our discussion on the problem of egg adaptation.

  1. 193-197: how did this information directly impact the HTA, and later on, the Guidelines for vaccination in Italy?

Thank you for your observation, we have shortened the sentence. The recommendations released by WHO have impact worldwide. Indeed, we did not further elaborate on their impact but we have left a brief reference to it.

  1. 201: it is unclear why SARS-CoV-2 is cited here, since there is no available vaccine yet, and the decision-making in regard to vaccination in regard will probably be influenced by dynamics very different from traditional HTA evaluation, once available. It is strongly encouraged to avoid references to topics that are unrelated, and thus unnecessary.

Thank you for your observation, we have removed the reference to SARS-CoV-2.

  1. Nature of submission: this article has been submitted as “Communication” (at the best of my knowlegde), but does not match the requirement of “preliminary, but significant, results” as defined by Vaccines, since this manuscripts mostly summarizes previously published works and policies, that are already in place

Thank you for your observation. We agree with you as the paper could be more properly classified as a perspective. We selected “communication” as we did not find anything else more appropriate. 

Language and editing:

  1. “keep ongoing”?.

Also following the advice of the Reviewer 3 we have eliminated the words “keep ongoing”.

  1. “to guiding”? To guide or guiding? For effective guidance? Authors should revise the title in its entirety.

Thank you for your suggestion. We have changed the title as follows “Strengthening the evidence-based approach to guiding effective influenza vaccination policies”.

  1. 27: allocation process of what? Of fundings? Of the vaccine to patients?

Thank you for your warning, we have rephrased that part as follows “the process of allocating resources to vaccination strategies

  1. VPD: please specify acronym

Thank you for your warning, we have indicated the full name “Vaccine Preventable Diseases”.

  1. Indeed? It is unclear what authors mean

Thank you for your warning, we have rephrased that part as followsand of the whole world”.

  1. check tabulation line 41

Thank you for your warning, we have checked and updated references.

  1. The choice of English translations for some words - while not depriced of sense - could be revised with some more commonly used terms to improve the understanding: i.e. repercussion (translated from “repercussioni”) vs impact; appropriateness (translated form “appropriatezza”) vs suitability.

Thank you for your warning, we have replaced repercussion with impact. In regard to appropriateness, we would like to maintain that term as different from suitability which mainly refers to the organizational context. We would also like to inform you that we have submitted the paper to language editing by an English mother tongue before the original submission.

  1. Furthermore, there are many sentences that the authors may want to rephrase in order to increase the comfort and understanding of the readers.

Thank you for your warning, we have already submitted the paper to language editing by an English mother tongue before the original submission.

Reviewer 3 Report

General

Authors communicate in their manuscript ‘Keep ongoing technological development: Strengthening the evidence-based approach to guiding effective influenza vaccination policies’ that an evidence-based approach regarding vaccination (meant is a system containing HTA) will lead to better outcomes. I like this topic as I think this approach to vaccination is understudied, however some work should be done before this manuscript can be published.

Title

Please shorten the title a bit; according to me the first part <Keep ongoing technological development:> could be missed.

Abstract

Background

I miss a bit the view of the patient; the communication is written as if HTA would be carried out – then the patient will trust the system, what is not consistent with reality; patient plays his own role in the adherence. Perhaps it makes a difference when vaccination is compulsory or voluntary.

Harmonize the Aim with the last sentence of the Introduction; they are different.

Introduction

39           please explain what might mean <VPDs>

44           you are aware of the problem of people who don’t want to be vaccinated, as you say <… vaccination coverage is low or suboptimal …>

45           i would like you to add also <life expectancy>; please choose your own words for expressing this.

92-95     the objectives here differ from the Abstract; please harmonize

HTA in Italian vaccination policy

105         HTA is not enough; in the very end the patient decides whether he/she will adhere or not. Please add.

HTA in Italian influenza vaccination policy

my biggest problem with this manuscript is that HTA itself is not enough to make a success of a certain vaccination, but that an organisation has to change the patient's adherence.

Author Response

We thank the Reviewer for the general comment and her/his suggestions that have given us the possibility to improve the quality of the manuscript. Hereafter, point-to-point answers have been reported.

Title

  1. Please shorten the title a bit; according to me the first part <Keep ongoing technological development:> could be missed.

We thank you for your advice. We have reduced the title according to your suggestion.

Abstract

  1. I miss a bit the view of the patient; the communication is written as if HTA would be carried out – then the patient will trust the system, what is not consistent with reality; patient plays his own role in the adherence. Perhaps it makes a difference when vaccination is compulsory or voluntary.

Thank you for your observation, that has given us the possibility to improve the quality of the abstract. In fact, we have included in the abstract a reference to the role of citizens and the need to support their empowerment and engagement in the decision-making process.

  1. Harmonize the Aim with the last sentence of the Introduction; they are different.

Thank you for your warning. We have harmonized the aim with the last sentence of the introduction. Please bear in mind that the first objective of the paper has been modified following a comment of Reviewer 2.

Introduction

  1. 39 please explain what might mean <VPDs>

Thank you for your warning, we have indicated the full name “Vaccine Preventable Diseases”.

  1. 44 you are aware of the problem of people who don’t want to be vaccinated, as you say <… vaccination coverage is low or suboptimal …>

Thank you for your observation. Based also on the suggestions of Reviewer 2, we have reduced our introduction and therefore that sentence has been removed. However, the following sentence addresses the topic "Nonetheless, despite international and national recommendations on influenza vaccination and the availability of a broad spectrum of safe, effective and cost-effective vaccines, vaccination coverage levels, both in Italy and in Europe, remain rather low”

  1. 45 i would like you to add also <life expectancy>; please choose your own words for expressing this.

Thank you for your advice. We have included the following sentence “Indeed, influenza imposes an important burden on health, the healthcare system and the society.“ Furthermore, we have also included data on DALYs attributable to influenza.

  1. 92-95 the objectives here differ from the Abstract; please harmonize

Thank you for your advice. We have harmonized the objectives with the contents of the abstract. Please bear in mind that the first objective of the paper has been modified following a comment of Reviewer 2.

HTA in Italian vaccination policy

  1. 105 HTA is not enough; in the very end the patient decides whether he/she will adhere or not. Please add.

Thank you for your suggestion. We have added the following sentence “However, the success of a vaccination strategy depends upon the actions of several stakeholders, citizens among them. In fact, in the very end, the citizen decides whether he/she will adhere or not at the vaccination”.

HTA in Italian influenza vaccination policy

  1. my biggest problem with this manuscript is that HTA itself is not enough to make a success of a certain vaccination, but that an organisation has to change the patient's adherence.

Thank you for your valuable comment. We agree with you but the topic would deserve an ad hoc paper. Nevertheless, at the end of section 2 we have included the following sentence “Therefore, citizens are expected to participate in the decision-making process through involvement in patient advocacy groups, advisory boards and HTA bodies [11]. They should also eventually be engaged and empowered in order to make a proper use of technologies”.

Round 2

Reviewer 2 Report

It is appreciable to read that the authors have thoroughly adressed the comments, by extensively modifying and rewriting entire parts of the manuscript.

The manuscript, in its present form, represent an interesting read on vaccination decision-making in Italy.

Reviewer 3 Report

authors incorporated all comments i had, so i suggest acceptance now